# Male Clinical Parameters (Age, Stature, Weight, Body Mass Index, Smoking History, Alcohol Consumption) Bear Minimal Relationship to the Level of Sperm DNA Fragmentation

**DOI:** 10.3390/jpm13050759

**Published:** 2023-04-28

**Authors:** Shiao Chuan Chua, Steven John Yovich, Peter Michael Hinchliffe, John Lui Yovich

**Affiliations:** 1PIVET Medical Centre, Perth, WA 6007, Australia; schua@pivet.com.au (S.C.C.); syovich@pivet.com.au (S.J.Y.); apwin@pivet.com.au (P.M.H.); 2Hospital Shah Alam, Shah Alam 40000, Selangor, Malaysia; 3Faculty of Health Sciences, School of Medicine, Curtin University, Perth, WA 6102, Australia

**Keywords:** sperm DNA fragmentation (SDF), age effects, smoking effects, alcohol effects, stature, body mass index (BMI)

## Abstract

This retrospective cohort study reports on 1291 males who were the partners of women presenting with infertility requiring assisted reproduction and who had sperm DNA fragmentation (SDF) levels measured by the Halosperm test. These men provided clinical and biometric details which included their age, stature, weight, and body mass index (BMI). Of these men, 562 (43.5%) provided detailed historical records of their smoking and alcohol histories. The aim of this study was to determine whether any clinical and biometric parameters, or main lifestyle factors, had any influence on SDF. We found that the only clinical parameter with a direct correlation was that of advancing age (r = 0.064, *p* = 0.02), but none of the biometric parameters of stature, weight, or BMI showed any significant correlation. In respect to lifestyle, there were significant correlations with smoking history, but not in the way we expected. Our data showed significantly elevated SDF levels among non-smokers (*p* = 0.03) compared with smokers. We also found that, among the non-smokers, ex-smokers had higher SDF levels (*p* = 0.03). With respect to alcohol, consumers did not show any significant differences in SDF levels. These lifestyle findings did not show any significant relevance with respect to an SDF level of <15% or ≥15%. Furthermore, logistic regression analysis excluded age as a confounder in these lifestyle findings. It is therefore concluded that, apart from age, both clinical and lifestyle aspects have minimal relevance to SDF.

## 1. Introduction

Spermatogenesis occurs in seminiferous tubules and is accomplished in four sequential stages [1,2] in a process which takes about 72 days [3]. The first stage is spermatocytogenesis, which occurs in the basement membrane of the tubules where the spermatogonia undergo mitosis and generate B-type spermatogonia which then further divide by mitosis and differentiate into primary and secondary spermatocytes [1,2]. The second stage is spermatidogenesis, which includes meiotic division of spermatocytes, producing spermatids. Spermiogenesis is the third stage and involves transformation of round spermatids into elongated spermatids. During this transformation, remodelling of the sperm chromatin occurs where the nuclear histones are replaced by transition proteins and thereafter by protamines [1,2,4]. Sperm DNA strand breaks occur physiologically during this maturation process; however, excessive sperm DNA fragmentation (SDF) may arise if there is an imbalance between the rate of breaks and repair processes [5]. The final step is spermiation, which involves the release of mature spermatozoa into the lumen of seminiferous tubules; these are initially nonmotile and incapable of fertilizing an oocyte [1,2,6]. Assisted by smooth muscle contractions as well as ciliary movements, spermatozoa are passively transported from the seminiferous tubules to the vasa efferentia of the epididymis [6]. Spermatozoa spend around 12 days within the epididymal caput and undergo biochemical maturation, which involves changes in the glycoproteins of the plasma membrane of the sperm head. Spermatozoa develop the capacity for fertilization once they reach the caudus epididymis [6], These processes of capacitation and fertilization can also currently be achieved by assisted reproductive techniques (ART) [7].

As a consequence of this relatively long spermatogenetic process, the sperm appear prone to DNA and chromatin damage at various stages. It may occur within the testis, the male genital ducts, or even after ejaculation [8]. Defective chromatin remodelling, abortive apoptosis and reactive oxidative stress (ROS) are the major causes of SDF [4,8]. Apoptosis is an essential physiological process to remove abnormal germ cells and to maintain the ratio of germ cells to Sertoli cells within the testis [9]. However, aberrant apoptosis could adversely affect either sperm production or DNA fragmentation [10], the latter being one of the markers of apoptosis found in ejaculated spermatozoa and which may arise excessively in men with infertility from incompetent spermatozoa [10]. Seminal oxidative stress and apoptosis are interlinked and comprise a unified pathogenic mechanism for sperm DNA damage [10,11]. Oxidative stress happens when there is an imbalance between ROS production and the antioxidant defence mechanism within the body. Spermatozoa are susceptible to ROS, especially to lipid peroxidation, due to the presence of a large proportion of polyunsaturated fatty acids [12] and their lack of cytoplasmic enzyme systems [13]. Increased ROS levels in the semen with idiopathic male infertility has also been shown to negatively impact sperm function [14], and can be generated by a variety of endogenous (spermatozoa, immature germ cells, leucocytes, varicocele) and exogenous sources (clinical, environmental, and lifestyle risk factors) [15]. Smoking has been demonstrated as a known risk factor which can negatively affect both sperm quality and function [11,16,17,18,19] by elevated SDF [20,21]. The mechanism appears to be that cigarette smoking creates oxidative stress either by generating an excessive amount of ROS or by lowering the antioxidant capacity within seminal plasma [22]. Despite these reports, some studies have demonstrated that smoking does not appear to have any impact on sperm quality [23,24,25,26,27]. 

There is also ongoing debate concerning the impact of alcohol intake on semen quality, with previous studies showing inconsistent results. The majority of studies have found no association between alcohol intake and semen quality [24,27,28], whereas some found alcohol consumption can generate a negative impact on semen quality [29,30]. Furthermore, higher rates of sperm DNA fragmentation and chromatin decondensation [16,31] have also been observed among heavy drinkers. A meta-analysis which analyzed data from 18 cross-sectional studies concluded that semen quality did not seem to be made worse by occasional alcohol intake, and indeed, the authors observed even better sperm motility in occasional drinkers than those who never drink; whereas both semen volume and sperm morphology were negatively affected by daily consumption [32].

Likewise, the effect of body mass index (BMI) on SDF still remain controversial. Men with obesity were demonstrated to have higher intestinal permeability, metabolic endotoxemia, and sperm DNA oxidative damage, which together act to compromise the integrity of the sperm DNA in comparison to men who have normal body weight [33,34]. In contrast, a meta-analysis and a comprehensive review involving 8255 participants revealed inadequate evidence to establish a positive correlation between BMI and SDF [35]. 

The contradictions among studies on the consequences of smoking, alcohol consumption, and BMI concerning sperm quality and DNA integrity encouraged us to explore correlations between the relevance of clinical features: notably, stature, weight, and BMI, along with lifestyle factors: namely, smoking history and alcohol consumption. The influence of these parameters on SDF levels is assessed by the application of the sperm chromatin dispersion (SCD) test, i.e., the Halosperm test. 

## 2. Materials and Methods

### 2.1. Methodology

The Halosperm test was conducted on 2624 men attending PIVET Medical Centre (PIVET) between 1 March 2013 and 30 September 2022, the details of which have been fully described in an earlier publication [36], which examined DNA fragmentation index (DFI) and its correlation with semen analysis profiles. This retrospective study was conducted on 1291 of those men who were the partners (husbands and de facto husbands) of women presenting with infertility who also had stature (height), weight, and BMI measurements recorded, along with age recorded at the time of the test. A further study was conducted on 562 of these men who also had lifestyle aspects recorded, namely smoking and alcohol histories, which was introduced increasingly from 2018. These groups are shown in the Flowsheet (Figure 1).

Biometric parameters (height, weight, and BMI) were obtained at the initial presentation, being conducted by the Clinic Nurse prior to the formal consultation and clinical examination by the Consultant Clinician. Height was measured in meters (m) using a stadiometer fixed to a wall, and body weight was measured in kilograms (kg) with a digital scale (Tanita BC–545N, Kewdale, WA, Australia). Their BMI was calculated and recorded on the history clerking sheet, and thereafter transferred into the electronic database (Filemaker Pro Database). 

Both the smoking history and alcohol history were self-reported on the health and history questionnaire. The smoking history was reported as current and past history, whilst alcohol was reported as either never drinking or consumption on a daily, weekly, monthly, or only occasional basis. In this preliminary questionnaire, the units/amount and type of alcohol were not sought. The population with adequate details of their smoking and alcohol histories numbered *n* = 562 samples comprising 43.5% of the total population. 

There is no single specific cut-off value for DFI which has been unanimously agreed at this juncture. At PIVET, a threshold DFI of 15% is considered to be relevant for the indication of ICSI [37], as higher DFIs negatively correlated with blastulation and pregnancy rates [38] rather than DFI levels of ≥20% or ≥30% as reported elsewhere [21,39,40]. For this study, we therefore classified the SDF grouping into <15% and ≥15%. 

### 2.2. Laboratory Procedures

Semen samples were collected into a sterile plastic container by masturbation and labeled with the date and time of collection. All seminal fluid examinations were performed by the in-house andrology laboratory, where samples were maintained at room temperature until complete liquefaction. Semen analysis was performed according to the WHO standard 2010 [41]. Macroscopic examination including liquefaction, semen viscosity, semen volume, and pH. Microscopic examination assessed the sperm concentration, morphology, and motility which was sub-classified into progressive and non-progressive or non-motile. The sperm DNA fragmentation index (DFI) was determined with the SCD test using Halosperm^®^ G2 Kit (Parque Cientifico de Madrid, Spain), patented by Halotech.

#### Principle of Sperm Chromatin Dispersion Test—Halosperm^®^ G2

Our previous report has described the principle of the Halosperm test [36]. It is based on a controlled DNA denaturation process to facilitate the subsequent removal of the proteins contained in each spermatozoa. Intact unfixed sperm (fresh, frozen or unthawed, diluted samples) were immersed in an inert agarose microgel on a pre-treated slide. An initial acid treatment denatures the DNA in fragmented DNA sperm cells. Thereafter, the lysis solution was applied to remove most of the nuclear proteins. Sperm heads with massive loops of elongated DNA strands emerging from the central core signify the absence of DNA breakage and show a large to medium dispersion halo. The nucleoids from sperm with fragmented DNA either do not show a dispersion halo, or only a small halo [42].

The images of halos generated were highly contrasted and a total of 200 sperm cells were assessed under bright field microscopy. Those cells which did not exhibit a clear tail were not included in the sperm count for DNA fragmentation.

The percentage of sperm with fragmented DNA was calculated as below:$$\mathrm{DFI}\;(\%)=\frac{\mathrm{Fragmented}\;\mathrm{and}\;\mathrm{degraded}\;\mathrm{sperm}\;\mathrm{cells}\;}{\mathrm{Total}\;\mathrm{number}\;\mathrm{of}\;\mathrm{sperm}\;\mathrm{cells}\;\mathrm{counted}}\;\times \;100$$

### 2.3. Validation and Quality Control

The Halosperm test results of DFI were compared with controls and replicated by separate technicians (embryologists). If the values exceeded a permissible range of 25%, the test was repeated. The sensitivity and specificity of the test is 93% [42].

The Halosperm test has a quality control assessment from FertAid Pty Ltd. (Newcastle, NSW, Australia), who provide quality assurance and training in the Reproductive Sciences, and it monitors many aspects of the embryology and andrology aspects within the PIVET laboratory.

### 2.4. Statistical Analysis

All statistical analyses were performed using SPSS software (version 26.0, SPSS Inc.). We analyzed the numerical data using the Student’s *t* test or One-way ANOVA to compare the means and presented results as the mean value ± standard deviation. Correlation analysis was performed using bivariate Pearson analysis. The Chi-Square test or Fisher’s Exact test was used to determine the relationship between the two nominal (lifestyle) variables. Multivariate logistic regression was applied to determine confounding variables. Differences between the values were considered statistically significant when *p* < 0.05.

## 3. Results

### 3.1. Baseline Characteristics

We compared the clinical parameters based on SDF groups at <15% and ≥15%, as depicted in Table 1. The weight, total motility, and progressive motility were found to be significantly lower in the SDF ≥ 15% group.

### 3.2. Correlations between Clinical Parameters and Sperm DFI

The summary of correlation coefficients between sperm DFI and clinical parameters is shown in Table 2. Correlation analysis indicates that sperm DFI is positively associated with age (r = 0.064; *p* = 0.022), and is not correlated with stature, weight, or body mass index (BMI). The correlation graphs of four clinical parameters are shown in Figure 2.

### 3.3. Relationship between Lifestyle and Sperm DFI

In this aspect of the study, we specifically investigated the smoking and alcohol consumption status in relation to sperm DFI from those males who sought infertility treatment at our clinic. Non-smokers (*n* = 396) comprised 70.5%, whereas smokers (*n* = 166) were the minority at 29.5%. Most of the males (*n* = 481) consumed alcohol, comprising 85.6% of the group, and a minority (*n* = 81) comprising 14.4% of the group did not consume alcohol. These data are depicted in Figure 3 along with the proportion within each lifestyle sub-group.

Contrary to our expectations, we demonstrated there were significant differences between the non-smoker and smoker groups (*p* = 0.03). The mean DFI levels among the non-smoker group were considerably higher than those of the smoker group, although mean levels were <15% for both groups. Furthermore, compared to present smokers, ex-smokers showed an increased DFI (*p* = 0.03). No differences were observed between non-alcohol drinkers and their drinking counterparts (*p* = 0.09). Table 3 provides an overview of the findings with further information. We therefore further analyzed the non-smoker and ex-smoker groups with multivariate logistic regression but failed to determine any significant confounders. In particular, age proved not to be a confounding factor within the non-smoker and the ex-smoker groups. The mean age for the smoker group was 37.17 ± 6.92 years, while the mean age for non-smokers was 36.10 ± 6.21 years. Current smokers mean age was 36.06 ± 7.58 years, while former smokers mean age was 38.05 ± 6.24 years.

### 3.4. Subgroup Analysis of Various Lifestyle with Sperm DFI ≥ 15%

We compared each lifestyle group with DFI <15% and ≥15%, with the results shown in Table 4. No differences were observed among the smokers, ex-smoker, and alcohol consumers, nor their counterparts with respect to sperm DFI levels <15% and ≥15%.

## 4. Discussion

A significant declining trend in human fertility worldwide over the last six decades has been reported in the international literature [43]. This phenomenon raises important questions about modifiable environmental and lifestyle risk factors affecting human reproductive function [8], many of which are associated with the induction of oxidative stress within the reproductive tract [44]. Of those lifestyle risk factors, we have been interested to explore alcohol consumption, smoking, and increased BMI in correlation with SDF. Alcohol consumption and tobacco smoking are common lifestyle behaviors around the world. Globally, alcohol per-capita consumption increased from 5.9 L in 1990 to 6.5 L in 2017 and is expected to reach 7.6 L in 2030 [45]. In terms of smoking, Australian Bureau of Statistics showed one in ten people (10.7% or 1.9 million people) aged 18 years and above were current daily smokers, with men more likely than women to smoke daily (12.6% compared to 8.8%) in Australia in the years 2020–2021 [46]. Moreover, approximately 20% and 30% of individuals reported an increase in alcohol consumption and cigarette smoking, respectively, during the COVID-19 pandemic [47]. In this study, we also noticed that 29.5% of our patients were smokers and 85.6% of them consumed alcohol.

In this study, we were surprised that the SDF levels among non-smokers were significantly elevated compared with smokers; and no clear explanation can be offered other than the potential limitation related to the mismatched numbers between the two study groups. Interestingly, one double-blinded experimental study involving 117 infertile men demonstrated that smoking has a negative impact on intracellular antioxidant enzymes, however, that effect does not increase oxidative DNA damage [48]. Sperm DNA damage rises in proportion to the level of oxidative stress, and may be physiologically repaired in mammals, including humans, by the glycosidase DNA 8-Oxoguanine [49]. ROS does not always have an adverse effect, as a substantial body of research [50,51,52,53] also indicates that low concentrations of ROS are required for spermatozoa to develop normal fertilization capacities [12]. Spermatozoa are stimulated to undergo sperm capacitation, hyperactivation, acrosome reaction, and oocyte fusion when exposed to low concentrations of hydrogen peroxide (H_2_O_2_) [50,51,52,53]. Additionally, it has been demonstrated that nitric oxide and superoxide anions (O_2_^•−^) stimulate sperm capacitation and the acrosome response [54]. Numerous reports from the literature have also revealed that there was no relation between sperm motility [25,55,56], sperm morphology [55,56], or sperm concentration [25,56] and smoking. A large study carried out in China involving 1346 men also showed that smoking did not affect the semen parameters [57]. On the other hand, numerous reviews of the literature show that smoking has negative effects on sperm quality and function [11,16,17,18,19], as well as DNA integrity [20,21], which is mainly caused by oxidative stress [22].

We also found that there were no differences between alcohol drinkers and non- drinkers in terms of SDF levels. Nor were there any variations in DNA fragmentation across drinkers regardless of their drinking frequency. Our findings are consistent with a large retrospective study comprising 3976 semen samples obtained from the Andrology and Reproduction Laboratory across ten years in Argentina [24], and other studies which revealed no significant difference in semen quality [27,28]. A cross-sectional study among 347 Danish men revealed no significant dose–response association between semen profiles, including DFI and total alcohol intake [58]. In addition, a prospective study of 323 Italian males from infertile couples found that moderate drinkers (4–7 units of alcohol per week) had higher semen counts than non-drinkers, and drinkers of ≥8 units of alcohol per week were not negatively associated with other seminal variables including sperm motility [59]. Interestingly, a large study of 8344 healthy men from Europe and the USA also found no association between moderate alcohol consumption of any type (beer, wine, or liquor consumption) and semen quality [28]. Elsewhere, another study demonstrated that there was no significant differences in semen parameters between beer and wine drinkers [60]. While it is widely accepted that ethanol and its metabolized products are cell-toxic [61], it should also be mindful that wine or beer contain natural flavonoids and polyphenols such as resveratrol (RSV) or xanthohumol, which have been demonstrated to have antioxidant and anti-inflammatory properties as well as cell-protective potential [60,61]. Furthermore, RSV was particularly effective in preserving sperm chromatin texture [62] and demonstrated therapeutic and protective effects on spermatozoa [63].

Adipose tissue depots can enhance the pathway of oxidative stress via mitochondrial and peroxisomal fatty acid oxidation, subsequent to a pro-inflammatory state, which can affect reproductive potential and sperm function [64]. Several studies found that obesity was negatively associated with sperm function, sperm quality, and increased SDF [65,66,67]. Nonetheless, our study revealed there was no correlation between stature, weight, BMI, and SDF. This is in line with a prospective population-based study, the LIFE study, which analyzed 501 couples and demonstrated that neither BMI nor waist circumference had any discernible impact on semen concentration, motility, vitality, morphology, nor the SDF [68]. A meta-analysis which analyzed fourteen studies comprising 8255 participants also concluded that there was insufficient data to demonstrate a positive association between BMI and SDF [35].

Aging was observed to have a direct correlation with SDF in our study, which is consistent with a cross-sectional study that included semen samples from 2178 men and our previous study [36,69]. Aging is associated with both damage to the DNA of the nucleus and of the mitochondria, as well as a decrease in sperm motility, which are mainly caused by oxidative stress from ROS [69]. Age-related ROS accumulation leads to increased oxidative stress, which in turn triggers lipid peroxidation and further ROS production [70].

In addition to cigarette smoking, alcohol intake, and obesity, there is a vast array of risk factors that could potentially affect sperm quality. From a clinical standpoint, clinicians should elicit a comprehensive history of medical illness, recreational drug usage, psychological stress, diet, caffeine consumption, and evaluation of testicular size, genitourinary tract infections, as well as the presence of varicoceles when correlating with the semen profiles and SDF results [71,72,73]. Future studies should explore the possibilities of these other confounders influencing the SDF results.

## 5. Strengths and Limitations of the Study

There are a few study limitations worth mentioning: namely, the retrospective design and the incomplete data set of smoking and alcohol status on the males, and only 43.5% (562 out of 1291) of men being analysed. We are unable to completely rule out the possibility of selection bias or lingering confounders. In terms of alcohol consumption, the only information recorded from our males was the frequency, and not the actual amount nor the type of alcohol consumed. However, this measure (glasses/day or drinks/day) has been previously applied in other studies mentioned in the literature. Any future studies should include these details of alcohol consumption.

The strengths within our study relate to the fact that we have at least attempted to include the male in infertility assessment, beyond semen analysis, and believe this may eventually improve the prognosis for infertility management. Our next study in this respect will focus on ART outcomes related to the SDF level and whether the decision to apply ICSI may be more precisely guided. Our sample size is also relatively large for this analysis. The application of multivariate logistic regression analysis is a further strength in attempting the identification of confounding variables, in particular the influence of male age in deciphering the data concerning smokers.

## 6. Conclusions

Our study did not show a clear association between alcohol consumption, BMI, and DFI. Additionally, beyond our expectation, smokers were found to have significantly lower DFI compared to non-smokers. However, despite being ruled out as a potential confounder, we nevertheless do not encourage smoking in couples attempting to conceive. In particular, smoking and passive smoking may have adverse effects beyond the scope of these SDF studies. Males within infertile settings should be advised of the possible impact of daily lifestyle risk factors. As these lifestyle factors affecting male infertility are potentially modifiable, appropriate counselling and therapy intervention may be beneficial for couples within ART programs.

## Figures and Tables

**Figure 1 jpm-13-00759-f001:**
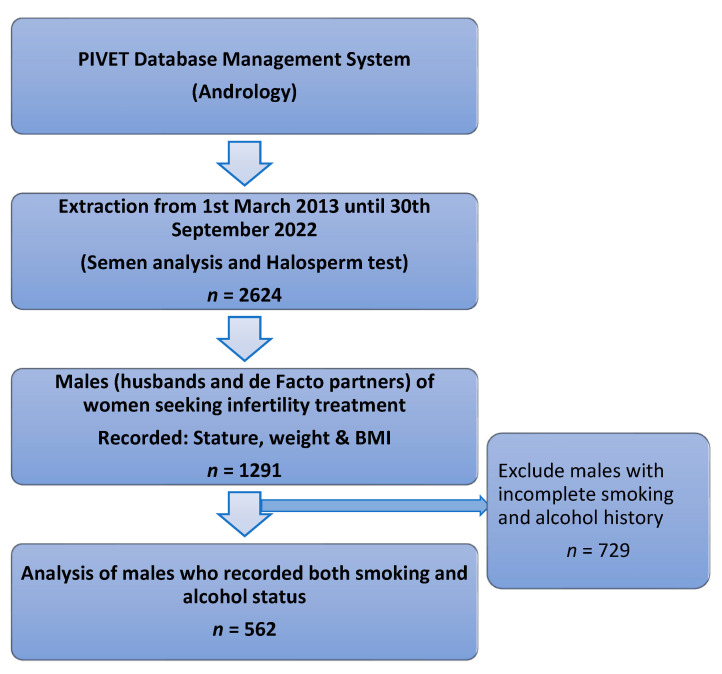
Flow chart of data extraction.

**Figure 2 jpm-13-00759-f002:**
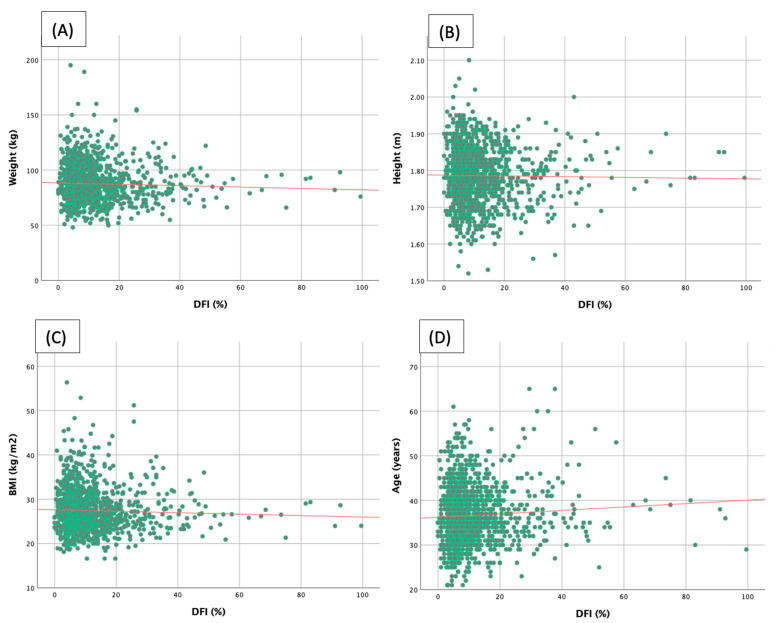
Correlation graphs between clinical parameters and DFI. Red lines indicate correlation trends; (**A**) weight, (**B**) height and (**C**) BMI show no significant correlation, (**D**) significant direct trend with age.

**Figure 3 jpm-13-00759-f003:**
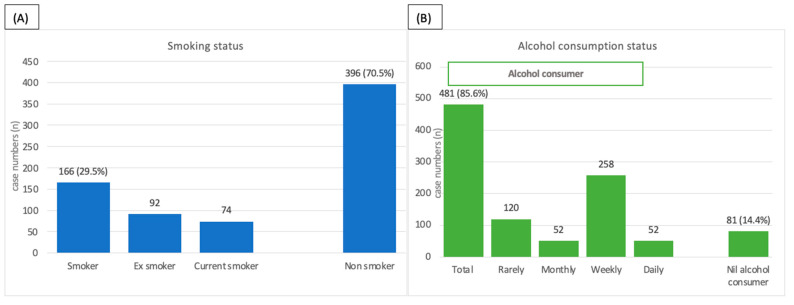
Lifestyle aspects of the males from infertility scenarios showing (**A**) the proportion of smokers and non-smokers, and (**B**) the proportion of alcohol and nil alcohol consumers along with sub-groupings.

**Table 1 jpm-13-00759-t001:** Comparison of mean ± SD of clinical parameters in SDF < 15% and ≥15% groups.

Clinical Parameters	Mean ± SD		*p*-Value
	SDF < 15% (*n* = 990)	SDF ≥ 15% (*n* = 301)	
Age (years)	36.49 ± 6.38	37.30 ± 6.84	0.06
Stature (m)	1.79 ± 0.08	1.78 ± 0.07	0.33
Weight (kg)	88.2 ± 16.72	85.84 ± 16.04	**0.03**
BMI (kg/m^2^)	27.56 ± 4.65	26.96 ± 4.56	0.05
SDF (%)	7.13 ± 3.39	26.25 ± 13.67	**<0.0001**
Sperm concentration (10^6^/mL)	65.25 ± 48.92	60.35 ± 50.67	0.13
Normal morphology (%)	8.04 ± 0.45	8.01 ± 0.27	0.27
Total motility (%)	42.37 ± 32.60	35.31 ± 31.71	**0.001**
Progressive motility (%)	39.51 ± 31.05	32.31 ± 29.73	**<0.001**
Seminal volume (mL)	3.45 ± 1.55	3.64 ± 1.75	0.08

**Table 2 jpm-13-00759-t002:** Correlation among clinical parameters and sperm DFI.

Clinical Parameters	Pearson Correlation (r)	*p*-Value
Age (years)	0.064	**0.022**
Stature (m)	−0.015	0.588
Weight (kg)	−0.041	0.139
BMI (kg/m^2^)	−0.038	0.168

**Table 3 jpm-13-00759-t003:** Comparison of various lifestyles and their DFI values (Mean ±  SD; *p*-value).

Lifestyle	SDF Levels Mean ± SD (n)	*p* Value	Odds Ratio (95% CI)	*p* Value
Non-smoker Smoker Current smoker Ex-smoker	$12.38\;\pm \;12.12\;\left(396\right)$ 10.06 $\pm \;9.15\;\left(166\right)$ 8.34 ± 5.94 (74) $11.45\;\pm \;10.92\;\left(92\right)$	**0.03** **^a^** **0.03** **^a^**	0.99 (0.96–1.02) 1.0 (0.94–1.06)	0.39 ^1^ 0.97 ^1^
Nil alcohol consumer Alcohol consumer Rarely Monthly Weekly Daily	$13.66\;\pm \;14.42\;\left(81\right)$ 11.37 $\pm \;10.75\;\left(481\right)$ 10.84 ± 9.87 (120) 10.64 ±$12.23\;\left(52\right)$ 11.88 $\pm \;11.31\;\left(258\right)$ 10.59 $\pm \;8.11\;\left(51\right)$	0.09 ^a^ 0.71 ^b^		

*p*-value was calculated by ^a^ Independent Student *t* test or ^b^ One way Anova test. ^1^ Age is not a confounding factor by multivariate logistic regression test.

**Table 4 jpm-13-00759-t004:** Lifestyle groupings for based on DFI levels <15% and ≥15%.

Lifestyle	DFI < 15%	DFI ≥ 15%	Relative Risk	*p* Value
Smoker Non-smoker Current smoker Ex-smoker	135/166 (81.3%) 293/396 (74.0%) 63/74 (85.1%) 72/92 (78.3%)	31/166 (18.7%) 103/396 (26.0%) 11/74 (14.9%) 20/92 (21.7%)	0.72 0.69	0.06 ^a^ 0.32 ^b^
Alcohol consumer Nil alcohol consumer	370/481 (76.9%) 58/81 (71.6%)	111/481 (23.1%) 23/81 (28.4%)	0.81	0.30 ^a^

*p* values calculated by ^a^ Chi squared test or ^b^ Fisher’s Exact test.

## Data Availability

Under the terms for Accreditation, the data in this study are not placed in a publicly accessible repository but can be sourced by correspondence with the Medical Director; corresponding author.

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
