# Peer review of "Male Clinical Parameters (Age, Stature, Weight, Body Mass Index, Smoking History, Alcohol Consumption) Bear Minimal Relationship to the Level of Sperm DNA Fragmentation"

_jpm, 2023, doi:10.3390/jpm13050759_

Round 1

Reviewer 1 Report

In this manuscript, the authors analyzed the sperm DNA fragmentation (SDF) levels calculated by Halosperm test. They considered for the study the age, stature, weight, and body mass index of the patients and the detailed historical recording of their smoking and alcohol histories. They found elevated SDF levels among non-smokers compared with smokers. Furthermore, they found among the non-smokers, ex-smokers had higher DFI levels (p=0.02). Finally, they found that alcohol consumers had more favorable, lower SDF levels compared to alcohol non–consumers.
In my opinion, the paper is well written, the data is convincing and the discussion is supported by the results. I have only minor corrections and comments.

Lines 49-63: add more studies regarding alcohol consumption and cigarette smoking correlated with sperm parameters evaluation.
Lines: 109-111: the DFI formula is not clear.
Lines 113-116 (Figure 2): if possible add the scale bar to the picture for a better understanding of the dimension of the halo.
Lines 231-232: add more data from the previous study correlated with this one.
Lines 240-242: add references supporting this data.

Author Response

Introduction       revised

References        updated

Lines 49-63       more studies added.

Lines 109-111     The DFI formula has been clarified.

Figure 2               Has been removed; it is published elsewhere and is cited.

Lines 231-232     Revised for clarity with relevant studies cited.  

Lines 240-242   additional references added.

Reviewer 2 Report

On March 13th 2023 the same authors have published in your journal a similar article about DNA fragmentation again, on the same number of patients – 2624. The investigation is performed in the same clinics. However, in the first article the number of patients on which DNA fragmentation assay is carried out is different from the one, cited in the present manuscript. Moreover, figure 2 in the manuscript is completely identical with figure 2 in the already published article, which I believe is not acceptable.

Regarding the current manuscript, its style is complete not understandable, some of the sentences are ambiguous. The methodology of the investigation is not correctly presented, it is unclear how many patients are actually are included. In the abstract it is mentioned that they are 1359, but on Figure 1 some of those are excluded. Furthermore, on the same figure it is completely unnecessary to present irrelevant exclusion criteria.

The statistical analysis and the discussion do not reflect the fact that the DNA fragmentation can be due to a combination of the investigated factors.

The authors divide the patients into 5 groups, depending on their percentage DFI. The distribution of smokers/non-smokers and alcohol/non-alcohol consumption parameters within the groups are shown on figure 5, but not the other investigated ones – age, height, BMI.

The argumentation in the discussion is not based on evidence and is not relevant. For example, the results showing better sperm quality in alcohol consuming patients are linked to the resveratrol in the red wine, but is not investigated what kind of alcohol the patients have been actually taking. Moreover, a significant part of the discussion is based on the assumption that the correlation shows causal relationship, but it is a well-known fact that it does prove such.

Some of the references are cited inappropriately.

Author Response

English attended by corresponding author

Introduction markedly upgraded

Citations extensively upgraded

Methodology upgraded with clearer explanation of derivation of male groups. Flowsheet  Figure1, has been upgraded. Identical Figure 2 has been removed, retaining citation only.  

DFI levels no longer presented in 5 groups, Accordingly, Figure 3 has been removed.

Figure 5 no longer requires any changes. It only refers to lifestyle aspects, namely smoking and alcohol.

Statistical evaluation concerning SDF revised

Discussion: Limitations regarding alcohol histories is acknowledged; limitations regarding potential causal relationships from correlations is also acknowledged and modified.

References extensively modified throughout the revised manuscript.

Reviewer 3 Report

Overall, the idea of the manuscript is interesting. Indeed is important to increases data to the conflicting data about the negative effects of smoking and drinking on sperm quality and function. However, the manuscript has some limitations thus authors should be more careful to taken conclusions.  

Authors should be more consistent in the design of tables and all should be more like table 1.

More suggestions above.

Abstract

Line 15-17. This sentence is quite confusing, please make it more simple and clear.

Line 26: “x-smokers had higher DFI levels (p=0.02).DFI is not previously defined.

Materials and methods

Line 68-70: Should be clearly indicated the percentage of men who are infertile (and the infertility causes, ex low sperm motility, morphology etc), and the percentage of men with normal semen parameters.

Line 77-81: If not used this metrics, please remove to avoid confusion.

Figure 1. First the design should be changed as it is difficult to read. Second the numbers must be revised. The males excluded as no having data for smoking or drinking are more than the global n.. it makes no sense. Which is your really n? I’m confused.

Results

Line 140-141: The authors should define which groups are control and SDF group.

Line 188-190: how authors can ensure that is not by change? The mean difference among groups is 2%, and as you say “Non-smokers (n=456) comprised 71.6% whereas smokers (n=184) were the 177 minority at 28.4%” and “ experience reveals that the positive controls (for higher 121 values of DFI) showed differences of  8.0%.. “ thus how can you guarantee this difference is causal?

The same for alcohol .

Thus carefull should be taken in the conclusions that authors state in abstract

“Our data actually showed significantly elevated SDF levels among non-smokers (p<0.05) compared with smokers, a finding which accords with some other studies. We also found that, among the non-smokers, ex-smokers had higher DFI levels (p=0.02). Furthermore, alcohol consumers had more favourable, lower SDF levels compared to alcohol non–consumers (p<0.05).” It may lead people to think that drinking and smoking is beneficial 

Author Response

English has been extensively adjusted by corresponding author.

Tables have been adjusted similar to Table 1.

Abstract: Extensively revised and DFI replaced by SDF

M&M: Revised for clarity

Lines 77-81 Metrics removed

Figure 1 Flowsheet extensively revised and clarified. Derivation of males into study groups clarified.

Line 140-141 SDF groupings revised/removed

Line 188-190 Limitations regarding the lower population of smokers does indeed affect the statistics, which we have now acknowledged. Similarly for alcohol rates.

Accordingly, the Abstract has been markedly revised and inferences concerning the positive aspects of smoking and alcohol have been heavily modified.

Round 2

Reviewer 2 Report

N/A